# Clinical Impact of CTLA-4 Single-Nucleotide Polymorphism in DLBCL Patients Treated with CAR-T Cell Therapy

**DOI:** 10.3390/curroncol32080425

**Published:** 2025-07-29

**Authors:** Katja Seipel, Inna Shaforostova, Henning Nilius, Ulrike Bacher, Thomas Pabst

**Affiliations:** 1Department for Biomedical Research, University of Bern, 3008 Bern, Switzerland; 2Department of Medical Oncology, Inselspital—Bern University Hospital, University of Bern, 3010 Bern, Switzerland; innaivanovna.shaforostova@insel.ch; 3Department of Clinical Chemistry, Inselspital—Bern University Hospital, University of Bern, 3010 Bern, Switzerland; henning.nilius@insel.ch; 4Department of Hematology, Inselspital—Bern University Hospital, University of Bern, 3010 Bern, Switzerland; veraulrike.bacher@insel.ch

**Keywords:** B-lymphocyte antigen CD19, cytotoxic T-lymphocyte-associated protein 4 (CTLA-4), single-nucleotide polymorphism (SNP), minor allele frequency (MAF), CAR-T cell therapy, FMC63-chimeric antigen receptor (FMC63-CAR), tisagenlecleucel (Kymriah©), axicabtagene ciloleucel (Yescarta©), lisocabtagene maraleucel (Breyanzi©)

## Abstract

Less than half of the patients with diffuse large B-cell lymphomas (DLBCLs) achieve durable remission with the current CD19-directed chimeric antigen receptor (CAR) T cell therapies. The molecular mechanisms underlying resistance to CAR-T cell therapy remain poorly understood. T cell activation is modulated by immune-checkpoint regulators. Genetic variants of immune-checkpoint regulators may have differential effects on T cell activation and response to CAR-T cell therapy. Cytotoxic T-lymphocyte-associated protein 4 (CTLA-4) is one of the inhibitory immune-checkpoint regulators. Here, we analyzed the treatment outcomes in DLBCL patients and discovered that *CTLA4* genetic variants may have clinical impact on the treatment outcome in DLBCL patients treated with the current CD19 CAR-T cell therapies.

## 1. Introduction

Chimeric antigen receptor CAR-T cell therapies targeting CD19 have greatly improved the prognosis for patients with relapsed/refractory large B-cell lymphoma (r/r DLBCL) [1,2,3,4,5,6]. The complete remission (CR) rates of 43–53% and durable remissions in up to 40% of these patients indicated a fundamental shift in the management of patients with r/r DLBCL. Historically, these patients had poor prognoses following salvage immuno-chemotherapy and/or autologous stem-cell transplantation. Despite the encouraging remission rates, 30 to 60% of patients experienced relapse after CAR T cell therapy [7,8].

Some clinical features and particular processes have been considered contributing to resistance in CAR T cell therapy: antigen escape, modulation of the tumor microenvironment and tumor burden [9,10,11,12]. However, the exact resistance mechanisms are still not fully understood [13]. Genetic variants of the CD19 target antigen may contribute to explaining differences in the response to CAR-T cell treatment [14]. In addition, genetic variants of immune-checkpoint regulators may also have an impact on T cell activation and response to CAR-T cell therapy. Immune-checkpoint receptors can stimulate T cell activity (CD27, CD28, CD40, CD122, CD134, CD137, GITR, ICOS) or depress T cell function (BTLA, CTLA4, LAG-3, PDCD1, TIM-3, TIGIT) [15]. Several *CTLA4* genetic variants have been associated with autoimmune disease including two single-nucleotide (SNP) variants located in the non-coding region (rs3087243, rs5742909) and one SNP in the coding region, rs231775 [16]. *CTLA4* rs231775 has also been associated with cancer risk either as predisposing or protective factor [17,18,19,20]. On the molecular level, rs231775 entails the substitution of threonine to alanine in the N-terminal signal peptide (T17A). This variation may affect the processing of CTLA-4 in the endoplasmic reticulum with less effective glycosylation and reduced expression of the CTLA-4 protein [21]. This, in turn, may downregulate T cell inhibition, resulting in reduced inhibitory signaling and enhanced T cell activation [22]. In the context of CAR-T cell therapy, such effects could promote CAR-T cell expansion, persistence, and antitumor activity, while also influencing the risk of immune-mediated toxicities. Experimental data demonstrate that CTLA-4 deficiency in CAR-T cells leads to increased proliferation, sustained CAR expression, and improved antitumor efficacy, primarily by allowing for unopposed CD28 signaling during the early expansion phase [23].

For *CTLA4* rs231775 the global minor allele frequency of 0.37 (ALFA) indicates a 14% prevalence of A17 homozygotes, 46% T17A heterozygotes, and 40% T17 homozygotes. Here, we investigated the prevalence of the *CTLA4* rs231775 allele in peripheral blood monocytes of DLBCL patients treated with FMC63 anti-CD19 CAR-T cell therapy. Clinical characteristics and treatment outcomes were analyzed according to the *CTLA4* genetic background.

## 2. Materials and Methods

### 2.1. Patients

This retrospective single-center study included 111 lymphoma patients treated with CAR T cell therapy between 9 January 2019 and 10 January 2024 at the University Hospital of Berne, Switzerland. CAR-T cell products included tisagenlecleucel (tisa-cel, Kymriah©), axicabtagene ciloleucel (axi-cel, Yescarta©) and lisocabtagene maraleucel (liso-cel, Breyanzi©, JCAR017). The study was approved by the Ethics Committee Berne, Switzerland (decision number 2025-00853, date of approval 24 April 2025).

### 2.2. Study Endpoints

The primary endpoint of the study was clinical outcome, which included relapse rate, progression-free survival (PFS), and overall survival (OS) in patients with and without CTLA4 rs231775. Secondary endpoints included the correlation of CAR T cell persistence and administered CAR T cell product (tisa-cel vs. axi-cel), as well as the correlation with toxicities. The American Society for Transplantation and Cellular Therapy (ASTCT) has established a set of consensus grading criteria that were applied to assess and grade two particular syndromes: cytokine release syndrome (CRS) and immune-effector cell-associated neurotoxicity syndrome (ICANS).

### 2.3. Statistical Analysis

Progression-free survival (PFS) was defined as the time interval from CAR T cell infusion to disease reappearance, mortality, or last follow-up. Overall survival (OS) was defined as the time interval from CAR T cell infusion to the date of death from any cause. Categorical variables were analyzed using the Chi-square test. Non-parametric data were assessed using the Kruskal–Wallis test. A multi-variate analysis was conducted using Cox proportional hazards models to identify prognostic factors. All statistical analyses were conducted using GraphPad Prism 8^®^ and R version 4.4.2 (2024).

### 2.4. CTLA4 Gene Analysis

Peripheral blood monocytes (PBMCs) were collected prior to CAR-T cell infusion, and genomic DNA was subsequently extracted from these cells. The amplification of DNA fragments encompassing exon 1 of the CTLA4 gene was achieved through the utilization of the FIREPol polymerase (Solis Biodyne, Tartu, Estonia) and gene-specific forward primer 5′-CTGAAGACCTGAACACCGCTCCCA-3′, along with the reverse primer 5′-CACCTC-CTCCATCTTCATGCTCCA-3′. Sanger sequencing was mandated for Micro-synth AG, located in Balgach, Switzerland.

## 3. Results

### 3.1. Prevalence of the CTLA4 A17 Allele in European DLBCL Patients

The sequence of the CTLA4 gene exon1 was determined in the peripheral blood of 111 lymphoma patients evaluated for CAR-T cell therapy at Inselspital Bern. A total of 50 patients (45%) carried two major alleles encoding CTLA4 T17 (T17hom). A total of 46 patients (41%) carried one minor allele with the single-nucleotide polymorphism rs231775 (T17Ahet), and 15 patients (14%) carried two minor alleles with SNP rs231775 (A17hom), indicating a minor allele frequency (MAF) of 0.32–0.38. In the Allele Frequency Aggregator project (ALFA), accessed on July 29, 2025, rs231775 was indicated with MAF of 0.37 (www.ncbi.nlm.nih.gov/snp/docs/gsr/alfa/, last accessed on 22 July 2025).

### 3.2. Baseline Clinical Characteristics of the DLBCL Patient Cohort

In total, 111 lymphoma patients who received CD19-targeted CAR-T cell therapies at Inselspital Bern were included in the study. Baseline clinical characteristics were analyzed for the entire cohort and for the three genetic subgroups with CTLA4 rs231775 encoding alanine or threonine at amino acid position 17 (CTLA4 A17hom, T17Ahet, or T17hom) (Table 1). All lymphoma patients received initially R-CHOP immuno-chemotherapy and one to six additional treatment lines prior to registration for CAR-T cell therapy. The median age at initial diagnosis and at the time of CAR-T cell infusion did not differ significantly among the A17hom, T17Ahet, and T17hom subgroups. The median time from diagnosis to CAR-T cell infusion was numerically longer in the A17hom and T17Ahet subgroups compared to the T17hom subgroup. The majority of patients were initially diagnosed with de novo diffuse large B-cell lymphoma (DLBCL) and disease stage IV according to Ann-Arbor classification system. Strikingly, within the A17hom subgroup most patients had transformed DLBCL, after initial follicular lymphoma (FL), and received high-dose chemotherapy (HDCT) followed by autologous stem-cell transplantation (ASCT) prior to CAR-T cell therapy. In the T17hom and T17Ahet subgroups, most patients were diagnosed with de novo DLBCL and a minority received HDCT/ASCT prior to CAR-T cell therapy.

### 3.3. Clinical Characteristics and CAR-T Cell Therapy

Clinical characteristics and details of CAR-T cell therapy were analyzed for the entire cohort and for the three genetic subgroups with *CTLA4* rs231775 encoding alanine or threonine at amino acid position 17 (CTLA4 A17hom, T17Ahet or T17hom) (Table 2). The majority of patients had a high risk score (IPI 4–5) with equal risk distribution in the three genetic subgroups. At the time of CAR-T cell infusion the majority of the A17hom patients were in partial response and a minority had progressive disease, while the majority of T17hom and T17Ahet patients had progressive disease, confirmed by Positron Emission Tomography/Computer Tomography (PET-CT). Accordingly, 50% of the T17hom patients received bridging chemotherapy in the time of CAR-T cell production, compared to 27% of the A17hom subgroup. All patients underwent lympho-depleting chemotherapy, comprising fludarabine and cyclophosphamide, five days prior to CAR-T cell infusion. The majority of patients received tisagenlecleucel (tisa-cel, Kymriah©) or axicabtagene ciloleucel (axi-cel, Yescarta©), and 6 patients received lisocabtagene maraleucel (liso-cel, Breyanzi©, JCAR017). Most of the patients in the A17hom subgroup received tisa-cel, while half of the T17hom patients received axi-cel (*p* = 0.001). Several factors may influence the outcome of CAR-T cell therapy, including patient characteristics and the specific CAR-T product used.

### 3.4. Treatment Outcomes—Univariate Analysis

The outcomes of the CAR-T cell treatments were analyzed for the entire cohort and for the three genetic subgroups with *CTLA4* rs231775 encoding alanine or threonine at amino acid position 17 (CTLA4 A17hom, T17Ahet or T17hom) (Figure 1, Table 3). The median (interquartile range [IQR]) follow-up time was 35 months. Cytokine release syndrome (CRS) occurred in 83% of patients a median two days after CAR-T cell infusion, with 4% severe CRS (Grade 3). Most CRS cases occurred in the T17Ahet and T17hom subgroups, with fewer in the A17hom subgroup (*p* = 0.006). Immune Effector Cell-Associated Neurotoxicity Syndrome (ICANS) a median 6 days after CAR-T cell infusion occurred in similar numbers and grades in all three genetic subgroups, inclusive cases of severe ICANS (Grad 3–4). Peak levels of inflammatory markers C-reactive protein (CRP), Interleukin-6 (IL-6) and serum ferritin were reached at a median of 3, 4 and 10 days after CAR-T cell infusion, respectively. Peak-levels of C-reactive protein (CRP) and ferritin were moderate in the A17hom subgroup and elevated in the T17Ahet and T17hom subgroups, with maximal CRP peak levels in the T17Ahet subgroup and maximal ferritin serum peak levels in the T17hom subgroup. Peak IL-6 levels diverged significantly in the three genetic subgroups, with low levels in the A17hom subgroup, high levels in the T17hom and very high levels in the T17Ahet subgroup (*p* = 0.005). Peak levels of CAR-T product in the plasma were detected at a median 9 days after CAR-T cell infusion with median 4751 copies per microgram cell-free DNA. Circulating CAR-T DNA was more persistent in the A17hom patients with median 170 copies compared to median 70 copies in the T17hom subgroup 6 months after CAR-T cell infusion. Patients carrying the germline variant *CTLA4* A17 allele had a better treatment outcome. In the CTLA4 A17hom, T17Ahet, and T17hom genetic subgroups, the four-year progression-free survival rates were 77%, 59%, and 30% (*p* = 0.019), four-year overall survival was 79%, 41%, and 33% (*p* = 0.049), the relapse rates were 20%, 37%, and 56% (*p* = 0.025), and the death rates were 20%, 54%, and 52% (*p* = 0.049), respectively.

To address the potential bias introduced by different CAR-T product types, the treatment outcome was analyzed in the subset of 63 patients who had received tisa-cel (Figure 2, Table 4). Patients carrying the germline variant CTLA4 A17 allele had a better treatment outcome. In the CTLA4 A17hom, T17Ahet, and T17hom genetic subgroups, the four-year progression-free survival rates were 78%, 58%, and 33% (*p* = 0.13), four-year overall survival was 80%, 35%, and 27% (*p* = 0.08), the relapse rates were 20%, 41%, and 58% (*p* = 0.16), and the death rates were 33%, 59%, 58% (*p* = 0.29), respectively. The most significant difference in outcomes associated with the three genetic subgroups were the IL-6 peak levels, with low levels in the A17hom subgroup and high levels in the T17hom and the T17Ahet subgroup (*p* = 0.004), indicating an association of IL6 peak levels to the *CTLA4* genetic background, independent of the CAR-T product type. Accordingly, the CRS rates also differed significantly with lower rates in the A17hom subgroup (*p* = 0.049). Circulating CAR-T DNA, however, was equally prevalent and persistent in the three genetic subgroups with median 4200 and 228 copies 9 days and 6 months after CAR-T cell infusion, respectively, indicating that proliferation and persistence of CAR-T cells was not associated with the *CTLA4* genetic background.

### 3.5. Treatment Outcomes—Multivariate Analysis

In the multivariate analysis, the *CTLA4* minor allele A17 was a relevant indicator of favorable treatment outcome with a hazard ratio (HR) of 0.3 in both PFS and OS in the two CTLA4 homozygous groups (A17hom vs. T17hom) (Table 5). Age was a risk factor of overall survival, but not of progression-free survival. Sex was not a risk factor of treatment outcome. Moreover, the multivariate analysis revealed an HR of 0.89 in axi-cel vs. tisa-cel therapy, indicating similar efficacy of the two CAR-T cell products in this study.

## 4. Discussion

The germline *CTLA4* rs231775 polymorphism has been implicated in autoimmune diseases [16] and cancer risk. *CTLA4* rs231775 may act as a risk factor for colorectal and thyroid cancers, while conversely serving as a protective factor against breast, liver, pancreatic and malt lymphoma [17,18,19,20]. In our study on B-cell lymphoma, *CTLA4* rs231775 was not associated with lymphoma risk, with minor allele frequencies of 0.37 in the DLBCL cohort (sample size 111) and in the European population at large (ALFA sample size 509524, ncbi.nlm.nih.gov/snp/, access date 22 July 2025).

*CTLA4* rs2317875, however, may be a prognostic marker for CAR-T cell response in DLBCL patients. Preclinical studies have shown that genetic or pharmacologic inhibition of CTLA-4 signaling in T cells enhances CAR-T cell proliferation, effector function, and persistence. These findings support the hypothesis that germline variants leading to reduced CTLA-4 function may similarly improve CAR-T cell efficacy [22]. The A17 variant of CTLA4 rs2317875 was associated with reduced CTLA-4 surface expression, potentially leading to less inhibitory signaling and greater T cell activation capacity [23]. In this retrospective study, we present evidence for a favorable clinical outcome to CAR-T cell therapy in DLBCL patients carrying a germline *CTLA4* rs231775-encoding alanine at amino acid position 17 of the CTLA4 protein. In more detail, significant differences emerged in clinical outcome in CTLA4 A17hom vs. T17Ahet and T17hom carriers with four-year progression-free survival at 77%, 59%, and 30% (*p* = 0.019) and relapse rates of 20%, 37%, and 56% (*p* = 0.025). This distribution indicates a dose-dependent effect of *CTLA4* rs231775, as the progression-free survival and relapse rates of the heterozygous group (T17Ahet) were in between the two homozygous groups (A17hom and T17hom).

The germline *CTLA4* rs231775 may also have a protective effect regarding disease severity. The majority of CTLA4 A17hom carriers had an indolent follicular lymphoma transformed to DLBCL, while the majority of T17hom carriers had de novo DLBCL. Transformed lymphoma was associated with a favorable response to CAR-T cell therapy [24]. Moreover, at the time of CAR-T cell infusion, the majority of CTLA4 A17hom carriers were in remission, while the majority of T17hom carriers had progressive disease and required bridging chemotherapy before CAR-T cell infusion. The distribution of CAR-T cell products differed among the three genetic subgroups, although the selection of individual products was incidental. Most CTLA4 A17hom carriers received tisa-cel CAR-T cell therapy and showed favorable treatment outcomes, whereas half of the CTLA4 T17hom carriers received axi-cel CAR-T cell therapy, which was associated with increased CRS-related toxicity. A higher efficacy and toxicity had been ascribed to axi-cel compared to tisa-cel in a matched comparison study in r/r DLBCL [25]. We addressed the potential bias introduced by the different CAR-T product types by a stratified analysis of the treatment outcomes in the subset of patients treated with tisa-cel CAR-T cell therapy. Within the tisa-cel-treated patients there was a trend towards a better treatment outcome in patients carrying the *CTLA4* rs2317875, and a significant association of *CTLA4* genetic background with IL-6 peak levels and CRS rates, but not with CAR-T cell prevalence or persistence. Based on the multivariate analysis of the entire cohort *CTLA4* rs231775 was a relevant indicator of favorable treatment outcome in r/r DLBCL treated with anti-CD19 CAR-T cell therapy independent of disease status, age, sex, and type of CAR-T product applied.

The regulation of T cell function, including activation and exhaustion, is complex with an interplay of immune-checkpoint molecules CTLA4, LAG3, PD1 and many others. CTLA4 may directly regulate PD1–PDL1 interaction [26]. PD-1 and LAG-3 may interact to foster T cell exhaustion [27,28]. Polymorphic protein variants exist in most checkpoint regulators [29]. Depending on the specific combination of polymorphic protein variants and levels of surface expression of all the different components, the resulting pattern of T cell activation and exhaustion is individualized. *CTLA4* rs231775 may associate to reduced CTLA4 surface expression with induced and sustained activation of resident and infused T cells.

Based on the clinical impact of the *CTLA4* genetic variant there may be a potential benefit of CTLA-4 inhibitors within the CAR-T cell therapy of DLBCL patients. If the *CTLA4* rs231775 polymorphism affected the expression of the CTLA-4 protein, a reduction in CTLA-4 function may have favorable impact on CAR-T cell treatment outcome. CTLA4 function may be blocked by specific inhibitors. The CTLA-4 inhibitor ipilimumab has shown anti-tumor activity in some patients with DLBCL [30]. The combination of rituximab and ipilimumab had manageable toxicity and encouraging efficacy in R/R follicular lymphoma [31]. A future treatment strategy may be to directly delete the *CTLA4* gene on CAR-T cells to improve efficacy of CAR-T cell therapy [23].

## 5. Conclusions

The germline variant *CTLA4* rs231775 may have clinical impact on disease severity and treatment outcome in DLBCL, with an association of the *CTLA4* minor allele A17 to favorable outcome after CAR-T cell infusion. To confirm the impact of *CTLA4* rs231775 on CAR-T response, and evaluate the possible role of *CTLA4* in DLBCL pathogenesis, a larger retrospective study is required. Moreover, treatment outcome may be improved in the future, with next-generation CAR-T cell products lacking immune-checkpoint regulators, leading to a more sustained immune response.

## Figures and Tables

**Figure 1 curroncol-32-00425-f001:**
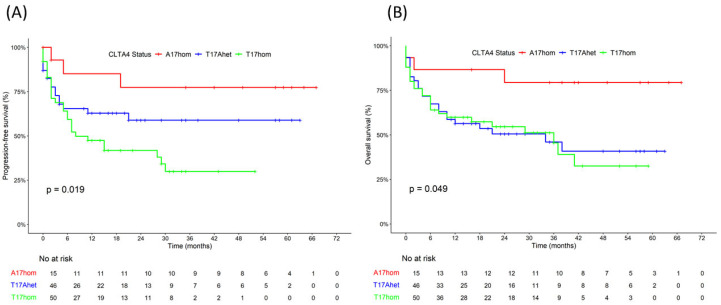
Survival outcomes of DLBCL patients after FMC63 CAR-T cell therapy according to *CTLA4* genetic background. (**A**) Progression-free survival (PFS) according to *CTLA4* gene polymorphism rs231775 encoding CTLA4 A17hom, T17Ahet, or T17hom. (**B**) Overall survival (OS) in genetic subgroups CTLA4 A17hom, T17Ahet, or T17hom.

**Figure 2 curroncol-32-00425-f002:**
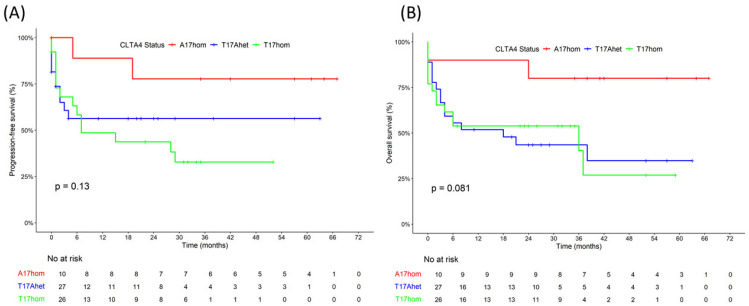
Survival outcomes of DLBCL patients after tisa-cel CAR-T cell therapy according to *CTLA4* genetic background. (**A**) Progression-free survival (PFS) according to *CTLA4* gene polymorphism rs231775 encoding CTLA4 A17hom, T17Ahet, or T17hom. (**B**) Overall survival (OS) in genetic subgroups CTLA4 A17hom, T17Ahet, or T17hom.

**Table 1 curroncol-32-00425-t001:** Baseline clinical parameters.

	All	CTLA4 A17hom	CTLA4 T17Ahet	CTLA4 T17hom	*p* *
**Patients,** *n* (%)	111 (100)	15 (14)	46 (41)	50 (45)	
Age at ID, years, median (range)	62 (24–79)	60 (43–75)	64 (34–78)	60 (24–79)	0.08 ^1^
M:F (ratio)	59:51 (1.2)	6:9 (0.67)	27:19 (1.7)	26:24 (1.4)	0.44 ^2^
Initial diagnosis	111 (100)				0.17 ^2^
DLBCL, *n* (%)	102 (92)	13 (92)	44 (96)	45 (90)	
de novo, *n* (%	66 (65)	4 (31)	31 (70)	31 (69)	
transformed, *n* (%)	36 (35)	9 (69)	13 (30)	14 (31)	
PMBCL, *n* (%)	2 (2)	0	0	2 (4)	
CLL/SLL, *n* (%)	1 (1)	1 (4)	0	0	
FL, *n* (%)	6 (5)	1 (4)	2 (4)	3 (6)	
Initial stage	91 (100)				0.36 ^2^
I, *n* (%)	5 (4)	0	2 (5)	2 (5)	
II, *n* (%)	16 (18)	1 (10)	7 (18)	8 (19)	
III, *n* (%)	17 (19)	5 (45)	3 (8)	9 (21)	
IV, *n* (%)	54 (59)	5 (45)	26 (69)	23 (55)	
Risk (IPI)	110 (100)				0.67 ^2^
Low risk (0–1), *n* (%)	3 (3)	1 (7)	1 (2)	1 (2)	
Low–intermediate risk (2), *n* (%)	8 (7)	1 (7)	3 (7)	4 (6)	
High–intermediate risk (3), *n*(%)	35 (32)	5 (33)	10 (22)	20 (40)	
High risk (4–5), *n* (%)	33 (30)	4 (27)	17 (40)	12 (24)	
Nd, *n* (%)	31 (28)	3 (20)	15 (33)	13 (26)	
Chemo-TX lines	111 (100)				0.26 ^2^
1, *n* (%)	5 (5)	1 (7)	3 (7)	1 (2)	
2, *n* (%)	39 (35)	3 (20)	19 (41)	17 (34)	
3, *n* (%)	58 (53)	8 (53)	20 (43)	30 (60)	
>3, *n* (%)	9 (7)	3 (20)	4 (9)	2 (4)	
Radiotherapy, *n* (%)	47 (42)	7 (47)	19 (41)	21 (42)	0.93 ^2^
ASCT, *n* (%)	42 (38)	9 (60)	13 (28)	20 (40)	0.08 ^2^

CAR-T: chimeric antigen receptor T cell; ID: initial diagnosis; IPI: international prognostic index; DLBCL: diffuse large B-cell lymphoma; FL: follicular lymphoma; CLL: chronic lymphocytic leukemia; PMBCL: primary mediastinal B-cell lymphoma; ASCT: autologous stem-cell transplantation; TX: therapy; * univariate analysis ^1^: Kruskal–Wallis test; ^2^: Chi-square test.

**Table 2 curroncol-32-00425-t002:** Clinical characteristics and details of CAR-T cell therapies.

Parameter	All	CTLA4 A17hom	CTLA4 T17Ahet	CTLA4 T17hom	*p* *
**Patients *n*** (%)	111 (100)	15 (14)	46 (41)	50 (45)	
Age at CAR-T, years, median (range)	66 (25–82)	67 (44–77)	71 (35–80)	64 (25–82)	0.19 ^1^
Time ID to CAR-T, years, median (range)	4 (1–26)	7 (1–13)	7 (1–19)	4 (1–26)	0.37 ^1^
Remission Status pre CAR-T					0.10 ^2^
CR	8 (7)	1 (7)	2 (4)	5 (11)	
PR	37 (35)	8 (57)	13 (29)	16 (33)	
SD	5 (5)	2 (14)	1 (2)	2 (4)	
PD	57 (53)	3 (22)	29 (65)	25 (52)	
Bridging chemo-tx	45 (41)	4 (27)	16 (35)	25 (50)	0.16 ^2^
Bridging radio-tx	18 (16)	1 (7)	7 (15)	10 (20)	0.56 ^2^
LDH pre CAR-T (U/L) median	331	326	346	304	0.35 ^1^
(range)	(135–3949)	(177–609)	(170–2312)	(135–3949)	
Elevated LDH (>250 U/L), *n* (%)	67 (60)	8 (53)	33 (72)	26 (52)	0.12 ^2^
CAR-T cell product					**0.001** ^2^
Tisa-cel	63 (57)	10 (67)	27 (59)	26 (52)	
Axi-cel	42 (38)	2 (13)	16 (35)	24 (48)	
Liso-cel	6 (5)	3 (20)	3 (6)	0	

IPI: international prognostic index; CR: complete response; PR: partial response; SD: stable disease; PD: progressive disease; LDH: lactate dehydrogenase; cfDNA: cell free DNA; Tisa-cel: tisagenlecleucel; Axi-cel: axicabtagene ciloleucel; Liso-cel: lisocabtagene maraleucel; * univariate analysis. ^1^: Kruskal–Wallis test; ^2^: Chi-square test. Bold formatting for *p* values < 0.05

**Table 3 curroncol-32-00425-t003:** Clinical outcome after CAR T cell therapies.

Parameter	All	CTLA4 A17hom	CTLA4 T17Ahet	CTLA4 T17hom	*p* *
Patients, *n* (%)	111 (100)	15 (14)	46 (41)	50 (45)	
CRS, *n* (%)	92 (83)	7 (47)	42 (91)	43 (86)	**0.006 ^2^**
Grade 1	61 (55)	5 (33)	29 (63)	27 (54)	
Grade 2	27 (24)	2 (13)	12 (26)	13 (26)	
Grade 3	4 (4)	0	1 (2)	3 (6)	
Time to CRS, days, median (range)	2 (0–17)	2 (0–4)	2 (0–12)	2 (0–17)	0.31 ^1^
ICANS, *n* (%)	39 (36)	6 (40)	19 (41)	14 (29)	0.29 ^2^
Grade 1	12 (11)	4 (26)	6 (13)	2 (4)	
Grade 2	7 (6)	0	5 (11)	2 (4)	
Grade 3	14 (13)	1 (7)	6 (13)	7 (15)	
Grade 4	6 (6)	1 (7)	2 (4)	3 (6)	
Time to ICANS, days, median (range)	6 (0–15)	7 (6–11)	5 (0–15)	6 (2–12)	0.58 ^1^
CRP peak, mg/L median (range)	43 (2–328)	25 (4–98)	48 (3–288)	43 (2–328)	0.06 ^1^
Time to CRP peak, days, median (range)	3 (0–28)	3 (0–10)	3 (0–28)	3 (0–22)	0.62 ^1^
IL-6 peak, pg/mL, median	606	31	1189	719	**0.005 ^1^**
(range)	(4–157,117)	(12–700)	(4–142,180)	(7–157,117)	
Time to IL-6 peak, days, median (range)	4 (0–39)	4 (0–7)	5 (1–39)	4 (0–21)	0.31^1^
Ferritin peak, ug/L, median	1210	572	1199	1367	**0.022 ^1^**
(range)	(99–13,393)	(161–4073)	(99–9791)	(190–13,393)	
Time to ferritin peak days, median (range)	10 (0–44)	9 (2–31)	10 (0–21)	10 (0–44)	0.62 ^1^
CAR T plasma peak, copies/µg cfDNA,	4751	4976	5662	4516	0.39 ^1^
median (range)	(30–218,000)	(1000–140,000)	(30–218,000)	(37–91,000)	
Time to CAR T peak, days, median (range)	9 (2–83)	10 (7–83)	9 (2–20)	9 (2–27)	0.72 ^1^
CAR T persistence 6 months, median	96	171	82	67	0.09 ^1^
(range)	(0–4071)	(65–4071)	(0–2069)	(0–2317)	
Best response after CAR T cell therapy					0.45 ^2^
CR, *n* (%)	61 (57)	11 (85)	25 (56)	26 (52)	
PR, *n* (%)	26 (24)	2 (15)	10 (22)	14 (28)	
SD, *n* (%)	6 (6)	0	3 (7)	3 (6)	
PD, *n* (%)	14 (13)	0	7 (15)	7 (14)	
Relapse, *n* (%)	48 (43)	3 (20)	17 (37)	28 (56)	**0.025 ^2^**
Death, *n* (%)	53 (48)	3 (20)	24 (54)	26 (52)	0.069 ^2^
Four-year PFS rate (%)	48	77	59	30	**0.019 ^3^**
Four-year OS rate (%)	45	79	41	33	**0.049 ^3^**

CRS: cytokine-release syndrome; ICANS: Immune Effector Cell-Associated Neurotoxicity Syndrome; PFS: progression-free survival; OS: overall survival; CR: complete response; PR: partial response; SD: stable disease; * univariate analysis. ^1^: Kruskal–Wallis test; ^2^: Chi-square test; ^3^: Log-rank test. Bold formatting for *p* values < 0.05

**Table 4 curroncol-32-00425-t004:** Clinical outcome after tisa-cel CAR T cell therapies.

Parameter	All Tisa-cel *n* (%)	CTLA4 A17hom	CTLA4 T17Ahet	CTLA4 T17hom	*p* *
Patients, *n* (%)	63 (100)	10 (16)	27 (44)	26 (41)	
CRS, *n* (%)	54 (86)	5 (50)	26 (96)	23 (88)	**0.049 ^2^**
Grade 1	35 (55)	3 (33)	18 (67)	14 (54)	
Grade 2	16 (25)	2 (20)	7 (26)	7 (27)	
Grade 3	3 (5)	0	1 (4)	2 (8)	
ICANS, *n* (%)	22 (35)	3 (33)	11 (41)	8 (31)	0.59 ^2^
Grade 1	7 (11)	2 (20)	3 (11)	2 (8)	
Grade 2	5 (8)	0	3 (11)	2 (8)	
Grade 3	6 (10)	0	4 (15)	2 (8)	
Grade 4	5 (8)	1 (10)	2 (7)	2 (8)	
CRP peak, mg/L median	43	19	48	44	0.16 ^1^
(range)	(2–328)	(4–98)	(3–288)	(2–328)	
IL-6 peak, pg/mL, median	480	26	677	696	**0.004 ^1^**
(range)	(4–157,117)	(12–700)	(4–142,180)	(7–157,117)	
Ferritin peak, µg/L, median	1113	494	948	1393	0.15 ^1^
(range)	(99–13,393)	(161–4073)	(99–9791)	(190–13,393)	
CAR T plasma peak, copies/µg cfDNA,	4212	4520	3157	4860	0.76 ^1^
median (range)	(30–218,000)	(1000–140,000)	(30–218,000)	(37–91,000)	
CAR T persistence 6 months, median	228	171	245	227	0.95 ^1^
(range)	(0–4071)	(65–4071)	(0–2069)	(0–2317)	
Best response after CAR T cell therapy					0.25 ^2^
CR, *n* (%)	34 (54)	8 (80)	14 (52)	12 (46)	
PR, *n* (%)	19 (30)	2 (20)	7 (26)	8 (31)	
SD, *n* (%)	4 (6)	0	3 (11)	1 (4)	
PD, *n* (%)	6 (10)	0	3 (11)	3 (12)	
Relapse, *n* (%)	28 (44)	2 (20)	11 (41)	15 (58)	0.156 ^2^
Death, *n* (%)	34 (48)	3 (33)	16 (59)	15(58)	0.29 ^2^
Four-year PFS rate (%)	52	78	54	30	0.13 ^3^
Four-year OS rate (%)	45	80	30	25	0.08 ^3^

CRS: cytokine-release syndrome; ICANS: Immune Effector Cell-Associated Neurotoxicity Syndrome; PFS: progression-free survival; OS: overall survival; CR: complete response; PR: partial response; SD: stable disease; * univariate analysis. ^1^: Kruskal–Wallis test; ^2^: Chi-square test; ^3^: Log-rank test. Bold formatting for *p* values < 0.05

**Table 5 curroncol-32-00425-t005:** Clinical outcome after CAR-T cell therapies—multivariate analysis.

Predictors	PFS		OS	
	HR	*p*-Value	HR	*p*-Value
CTLA4 A17hom vs. T17hom	0.31	0.06	0.27	**0.04**
CTLA4 A17hom vs. T17Ahet	0.51	0.30	0.34	0.09
Age > 65 vs. age < 65	0.82	0.56	1.83	**0.04**
Female vs. male	0.80	0.47	0.98	0.94
Axi-cel vs. Tisa-cel	0.89	0.70	0.70	0.23

hazard ratio (HR), complete remission (CR); progression free survival (PFS), overall survival (OS). Bold formatting for *p* values < 0.05

## Data Availability

Data available on request due to restrictions, privacy and ethics.

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
