# Peer review of "Clinical Impact of CTLA-4 Single-Nucleotide Polymorphism in DLBCL Patients Treated with CAR-T Cell Therapy"

_curroncol, 2025, doi:10.3390/curroncol32080425_

Round 1
Reviewer 1 Report
Comments and Suggestions for Authors
This single-center retrospective study presents a novel investigation of the clinical impact of the CTLA4 rs231775 SNP germ-line variants in 111 patients with DLBCL undergoing CD19-directed CAR-T therapy. The authors report that patients homozygous for the A17 allele had significantly improved outcomes, including PFS, OS, relapse, and CRS incidence. They further show differences in CAR-T persistence and inflammatory markers by genotype, and multivariable modeling supports CTLA4 A17 as an independent factor associated with favorable outcome.
While the findings are intriguing, several methodological and interpretational issues should be addressed to strengthen the validity and impact of the conclusions.
- Clarify and strengthen the biological rationale: The introduction acknowledges CTLA-4’s role as an immune checkpoint but falls short of articulating a mechanistic hypothesis explaining how rs231775 may influence CAR-T response. The T17A substitution has been associated with altered glycosylation and reduced CTLA-4 surface expression, but how this modulates endogenous versus CAR-modified T cell activation, persistence, or toxicity remains speculative. A clearer mechanistic framework would improve the coherence of the study. This should also be discussed in the discussion.
- Address potential confounding by CAR-T product type: A17hom patients disproportionately received Tisa-cel, which is known to have lower CRS rates and slower expansion compared to axi-cel. This may partially explain the observed differences in CRS, IL-6 levels, and CAR-T persistence. I would therefore be hesitant of reporting these without controlling for product or performing stratified analyses.
- Table 4: Clarify what is “CR within 6 months”. When was it measured, before or after CAR-T. If post-CAR-T, it should be removed from the model.
- Lack of functional validation: The study does not provide data and does not discuss how the rs231775 genotype affects CTLA-4 expression or function in CAR-T or endogenous lymphocytes. acknowledging this limitation more directly is important.
- Terminology: The manuscript refers to neurotoxicity as "CRES", but ICANS (immune effector cell-associated neurotoxicity syndrome) is the accepted term per ASTCT guidelines, which the authors cite. Terminology should be updated throughout.
- Clarify cohort description and re-order baseline tables: The results section quickly jumps into genotype subgroups without presenting a clear, unified description of the cohort. Consider beginning with a summary of the overall population, then comparing subgroups. Also, Table 1 would benefit from reordering to present age, sex, disease stage, and subtype before treatment history.
- Median follow-up time: Median follow up with IQR should be reported. eTime-to-event results (e.g., 4-year PFS and OS) are difficult to interpret without knowing the median follow-up. If the follow-up is substantially shorter, consider reporting 2-year outcomes or including a Kaplan-Meier risk table to indicate numbers at risk.
- Several tables list “univariable analysis” without specifying the test used (e.g., Fisher's exact, chi-square, t-test). Please include the specific test for each variable.
- Lines 204–207 describe rs231775 as a risk factor for some cancers and protective in others. This claim needs proper referencing, or alternatively should be removed if based on secondary synthesis.
- The manuscript would benefit from editing for clarity and conciseness. For example, Line 69: “before and after treatment” is repeated unnecessarily. Other examples of redundancy or awkward phrasing are scattered throughout.
- Abbreviations should be defined at first use and used consistently.
Comments on the Quality of English Language
Should be improved
Author Response
Reviewer 1. curroncol-3709465
Comments and Suggestions for Authors
This single-center retrospective study presents a novel investigation of the clinical impact of the CTLA4 rs231775 SNP germ-line variants in 111 patients with DLBCL undergoing CD19-directed CAR-T therapy. The authors report that patients homozygous for the A17 allele had significantly improved outcomes, including PFS, OS, relapse, and CRS incidence. They further show differences in CAR-T persistence and inflammatory markers by genotype, and multivariable modeling supports CTLA4 A17 as an independent factor associated with favorable outcome.
While the findings are intriguing, several methodological and interpretational issues should be addressed to strengthen the validity and impact of the conclusions.
Response: We thank the reviewer for the detailed critical comments. We have addressed all the issues. We believe that the manuscript has been substantially improved.
1) Clarify and strengthen the biological rationale: The introduction acknowledges CTLA-4’s role as an immune checkpoint but falls short of articulating a mechanistic hypothesis explaining how rs231775 may influence CAR-T response. The T17A substitution has been associated with altered glycosylation and reduced CTLA-4 surface expression, but how this modulates endogenous versus CAR-modified T cell activation, persistence, or toxicity remains speculative. A clearer mechanistic framework would improve the coherence of the study. This should also be discussed in the discussion.
Response: We have added a paragraph on the mechanistic framework to the introduction and discussion. We hypothesize that a reduced CTLA4 surface expression may upregulate both CAR-T cell activation and endogenous T-cell activation. The A17 (A allele) variant has been associated with reduced CTLA-4 surface expression, potentially leading to less inhibitory signaling and greater T cell activation capacity. In the context of CAR-T cell therapy, higher T-cell activity may enhance CAR-T cell expansion, persistence, and anti-tumor activity.
2) Address potential confounding by CAR-T product type: A17hom patients disproportionately received Tisa-cel, which is known to have lower CRS rates and slower expansion compared to axi-cel. This may partially explain the observed differences in CRS, IL-6 levels, and CAR-T persistence. I would therefore be hesitant of reporting these without controlling for product or performing stratified analyses.
Response: We have addressed the potential bias introduced by CAR-T product type by analyzing the treatment outcomes in the 63 patients who had received tisa-cel. The results are presented in a new paragraph within results section 3.4 including new Figure 2 and table 4. A new paragraph has been added to the discussion.
3) Table 4: Clarify what is “CR within 6 months”. When was it measured, before or after CAR-T. If post-CAR-T, it should be removed from the model.
Response: The CR status is assessed after CAR-T infusion. We believe that achieving an early complete response within 6 months is an important covariate, as the response to treatment and subsequent clinical outcomes is likely influenced by factors beyond the CTLA4 polymorphism alone. Including this covariate allows us to adjust for these additional, potentially unknown variables.
4) Lack of functional validation: The study does not provide data and does not discuss how the rs231775 genotype affects CTLA-4 expression or function in CAR-T or endogenous lymphocytes. acknowledging this limitation more directly is important.
Response: Indeed, this was a retrospective observational study and we did not personally address the mechanistic aspects of the rs231775 genotype. We have referred to results on CTLA4 functional aspects published by others.
5) Terminology: The manuscript refers to neurotoxicity as "CRES", but ICANS (immune effector cell-associated neurotoxicity syndrome) is the accepted term per ASTCT guidelines, which the authors cite. Terminology should be updated throughout.
Response: CRES was changed to ICANS.
6) Clarify cohort description and re-order baseline tables: The results section quickly jumps into genotype subgroups without presenting a clear, unified description of the cohort. Consider beginning with a summary of the overall population, then comparing subgroups. Also, Table 1 would benefit from reordering to present age, sex, disease stage, and subtype before treatment history.
Response: We have amended the results section 3.2. and reorganized table 1.
7) Median follow-up time: Median follow up with IQR should be reported. eTime-to-event results (e.g., 4-year PFS and OS) are difficult to interpret without knowing the median follow-up. If the follow-up is substantially shorter, consider reporting 2-year outcomes or including a Kaplan-Meier risk table to indicate numbers at risk.
Response: We have added a sentence on the median follow up with IQR in the report and included a Kaplan-Meier risk table in the survival analysis to indicate numbers at risk.
8) Several tables list “univariable analysis” without specifying the test used (e.g., Fisher's exact, chi-square, t-test). Please include the specific test for each variable.
Response: Categorical data in tables were analyzed by chi-square test, non-parametric data by Kruskal-Wallis test and survival data by log-rank test. This has been stated in the Materials and Methods section. We have included the information of the specific test for each variable in all tables.
9) Lines 204–207 describe rs231775 as a risk factor for some cancers and protective in others. This claim needs proper referencing, or alternatively should be removed if based on secondary synthesis.
Response: References have been added.
10) The manuscript would benefit from editing for clarity and conciseness. For example, Line 69: “before and after treatment” is repeated unnecessarily. Other examples of redundancy or awkward phrasing are scattered throughout.
Response: We have carefully checked and revised the manuscript accordingly.
11) Abbreviations should be defined at first use and used consistently.
Response: We have carefully reviewed the manuscript to ensure consistent use of all abbreviations and have verified that each abbreviation is clearly defined at its first occurrence.
Submission Date 03 June 2025
Date of this review 23 Jun 2025 12:35:33
Date of revision: 11 July 2025

Reviewer 2 Report
Comments and Suggestions for Authors
I think this is a solid hypothesis-driven paper which boasts of sound study design and presentation of data. The association of the genetic variants of CTLA-4 and DLBCL progression remains less clear compared to other cancers (e.g. colorectal, as the authors point out). To ask how the presence of genetic variants supports or hinders CAR T therapy makes for an intriguing prospect as well.
I did find the discussion to be a bit lacking and addressing two points would strengthen this analysis considerably, in my opinion:
- CTLA-4 regulates PD-1 levels (https://doi.org/10.15252/embj.2022111556), but it is not clear if there are differential mechanisms or efficacy by which these genetic variants of CTLA-4 may do so. I ask that the authors comment on this since such knowledge may inform the kind of checkpoint blockade/mAb regimens given to patients.
- Recent work has shown that CAR T therapies may actually work better in lymphoreplete patients. Consider: https://doi.org/10.1016/j.omto.2017.12.003,
- 10.1200/JCO.2015.64.5929, https://doi.org/10.1016/j.ymthe.2023.07.015 Since CTLA-4 may regulate the early expansion phase of T cells, can the authors speculate if/how these genetic variants may impact CAR T persistence?
- Finally, are these CTLA-4 genetic variants associated with specific HLA alleles? If so, a table showing as much would be useful and an important guide in the clinic.
Author Response
Reviewer 2 curroncol-3709465
Comments and Suggestions for Authors
I think this is a solid hypothesis-driven paper which boasts of sound study design and presentation of data. The association of the genetic variants of CTLA-4 and DLBCL progression remains less clear compared to other cancers (e.g. colorectal, as the authors point out). To ask how the presence of genetic variants supports or hinders CAR T therapy makes for an intriguing prospect as well.
I did find the discussion to be a bit lacking and addressing two points would strengthen this analysis considerably, in my opinion:
- CTLA-4 regulates PD-1 levels (https://doi.org/10.15252/embj.2022111556), but it is not clear if there are differential mechanisms or efficacy by which these genetic variants of CTLA-4 may do so. I ask that the authors comment on this since such knowledge may inform the kind of checkpoint blockade/mAb regimens given to patients.
Response: We have added a paragraph on the mechanistic interplay of CTLA4, LAG3 and PD1 which may vary depending on the polymorphic protein variants present, resulting in an individual pattern of T-cell activation and exhaustion. CTLA-4 regulates PD-1 pathway activity indirectly by modulating CD80 availability on antigen-presenting cells, thereby affecting PD-L1:PD-1 interactions and T cell inhibition. The rs231775 SNP may reduce CTLA-4 surface expression and function; however, there is currently no direct evidence that this or other CTLA-4 variants differentially influence PD-1 expression or signaling. Based on current evidence, CTLA-4 genetic variants such as rs231775 are not established biomarkers for selecting or predicting responses to PD-1/PD-L1 or CTLA-4 checkpoint blockade in DLBCL patients undergoing CD19-directed CAR-T cell therapy.
2. Recent work has shown that CAR T therapies may actually work better in lympho-replete patients. Consider https:// doi.org/10.1016/j.omto.2017.12.003, https://doi.org/10.1200/JCO.2015.64.5929, https://doi.org/10.1016/j.ymthe. 2023.07.015. Since CTLA-4 may regulate the early expansion phase of T cells, can the authors speculate if/how these genetic variants may impact CAR T persistence?
Response: In the current treatment settings lympho-depleting pre-conditioning is applied as this eliminates endogenous cytokine-responsive and immuno-suppressive cells, thereby promoting functional engraftment and persistence of transferred T cells. Even though CTLA4 rs231775 variants may enhance CAR T cell efficacy through effects on T cell activation and persistence, there is no clinical evidence to define their impact in lympho-replete patients or to support their use for patient selection or treatment modification.
- 1200/JCO.2015.64.5929, https://doi.org/10.1016/j.ymthe.2023.07.015 Since CTLA-4 may regulate the early expansion phase of T cells, can the authors speculate if/how these genetic variants may impact CAR T persistence?
Response: We have added a paragraph on the mechanistic framework to the intro-duction and discussion. We hypothesize that a reduced CTLA4 surface expression may upregulate both CAR-T cell activation and endogenous T-cell activation. The A17 (A allele) variant has been associated with reduced CTLA-4 surface expression, potentially leading to reduced inhibitory signaling and greater T cell activation capacity. Experimental data demonstrate that CTLA4 deficiency in CAR-T cells leads to increased proliferation, sustained CAR expression, and improved antitumor efficacy, primarily by permitting unopposed CD28 signaling during the early expansion phase. This suggests that individuals with CTLA4 variants associated with reduced inhibitory function, such as the A17 allele, may experience enhanced CAR T cell expansion and potentially greater persistence. However, excessive activation can also drive terminal differentiation and exhaustion, which may ultimately limit long-term persistence if not balanced appropriately.
Moreover, the type of CAR-T cell product may also impact treatment efficacy. We therefore addressed the potential bias introduced by CAR-T product type by analyzing the treatment outcomes in the 63 patients who had received tisa-cel. In this stratified analysis circulating CAR-T DNA was equally prevalent and persistent in the three genetic subgroups, indicating that expansion and persistence of CAR-T cells was not associated to the CTLA4 genetic background.
- Finally, are these CTLA-4 genetic variants associated with specific HLA alleles? If so, a table showing as much would be useful and an important guide in the clinic.
Response: CTLA4 gene polymorphisms are associated with several HLA alleles, in the context of autoimmune diseases. These associations suggest a complex interplay between CTLA4 and HLA genes in regulating immune responses and susceptibility to diseases like type 1 diabetes, Graves' disease, and rheumatoid arthritis. We agree that a possible association of CTLA4 with specific HLA alleles may be useful and important. Unfortunately, routine HLA testing is not performed in lymphoma patients undergoing CAR T-cell therapy at our center. Therefore, we are unable to address this question based on our current data.
Submission Date
03 June 2025
Date of this review
26 Jun 2025 18:14:18
Date of revision: 10 July 2025

Round 2
Reviewer 1 Report
Comments and Suggestions for Authors
All previously raised points have been addressed; however, one critical issue remains, which constitutes a major statistical flaw. In Table 5, response after CAR-T is included as a covariate in the model evaluating outcomes post-CAR-T. This approach is problematic for several reasons:
Including post-treatment response as a predictor introduces what is known as “immortal time bias” or “post-baseline variable bias.” By definition, response status is determined after CAR-T infusion and may be on the causal pathway between treatment and outcome. Incorporating variables that are determined after the start of follow-up violates the temporal ordering required for valid prediction or causal inference and can result in biased effect estimates. In particular, using post-treatment response as a covariate can lead to overestimation or underestimation of associations, as it conditions on a variable affected by the exposure and, potentially, by early events that influence both response and outcome.
In survival analyses, only baseline or pre-treatment covariates should be used for modeling post-treatment outcomes, unless appropriate methods for handling time-dependent covariates (such as landmark analysis or joint modeling) are employed. In the current model, including post-CAR-T response without such methods invalidates the analysis and its interpretation.
I strongly recommend that the analysis be removed or modified.
Author Response
Comments and Suggestions for Authors
All previously raised points have been addressed; however, one critical issue remains, which constitutes a major statistical flaw. In Table 5, response after CAR-T is included as a covariate in the model evaluating outcomes post-CAR-T. This approach is problematic for several reasons:
Including post-treatment response as a predictor introduces what is known as “immortal time bias” or “post-baseline variable bias.” By definition, response status is determined after CAR-T infusion and may be on the causal pathway between treatment and outcome. Incorporating variables that are determined after the start of follow-up violates the temporal ordering required for valid prediction or causal inference and can result in biased effect estimates. In particular, using post-treatment response as a covariate can lead to overestimation or underestimation of associations, as it conditions on a variable affected by the exposure and, potentially, by early events that influence both response and outcome.
In survival analyses, only baseline or pre-treatment covariates should be used for modeling post-treatment outcomes, unless appropriate methods for handling time-dependent covariates (such as landmark analysis or joint modeling) are employed. In the current model, including post-CAR-T response without such methods invalidates the analysis and its interpretation.
I strongly recommend that the analysis be removed or modified.
Response: We thank the reviewer for explaining the reason for the illegitimate inclusion of post treatment response in the multivariate analysis. To avoid any immortal time bias we have dismissed post treatment response in the new multivariate analysis, which resulted in modified HR and p values in the new table 5. Results section 3.5. has been modified, accordingly.
Submission Date
03 June 2025
Date of this review
18 Jul 2025 11:26:34
Date of revision 22 Juli 2025

Round 3
Reviewer 1 Report
Comments and Suggestions for Authors
The manuscript has been sufficient improved for publication.